# Association between Dietary Salt Intake and Progression in the Gastric Precancerous Process

**DOI:** 10.3390/cancers11040467

**Published:** 2019-04-03

**Authors:** Susan Thapa, Lori A. Fischbach, Robert Delongchamp, Mohammed F. Faramawi, Mohammed Orloff

**Affiliations:** 1Department of Epidemiology, College of Public Health, University of Arkansas for Medical Sciences, Little Rock, AR 72205, USA; sthapa@uams.edu (S.T.); RDelongchamp@uams.edu (R.D.); MElfaramawi@uams.edu (M.F.F.); MSOrloff@uams.edu (M.O.); 2Department of Biomedical Informatics, College of Medicine, University of Arkansas for Medical Sciences, Little Rock, AR 72205, USA; 3Winthrop P. Rockefeller Cancer Institute, University of Arkansas for Medical Sciences, Little Rock, AR 72205, USA

**Keywords:** salt intake, gastric cancer, atrophic gastritis, intestinal metaplasia, dysplasia

## Abstract

Gastric cancer is the third leading cause of cancer mortality worldwide. Studies investigating the effect of salt on gastric cancer have mainly used self-reported measures, which are not as accurate as sodium/creatinine ratios because individuals may not know the amount of salt in their food. Using data from a prospective cohort study, we investigated the effect of salt intake on progression to gastric precancerous lesions. Salt intake was estimated by urinary sodium/creatinine ratios, self-reported frequencies of adding salt to food, and total added table salt. We repeated the analyses among groups with and without *Helicobacter pylori* infection. We did not observe a positive association between salt intake, measured by urinary sodium/creatinine ratio, and overall progression in the gastric precancerous process (adjusted risk ratio (RR): 0.94; 95% confidence interval (CI) 0.76–1.15). We did observe an association between salt intake and increased risk for progression to dysplasia or gastric cancer overall (RR: 1.32; 95% CI: 0.96–1.81), especially among those who continued to have *H. pylori* infection at the five-month follow-up (adjusted RR: 1.53; 95% CI: 1.12–2.09), and among those who had persistent *H. pylori* infection over 12 years (adjusted RR: 1.49; 95% CI: 1.09–2.05). Salt intake may increase the risk of gastric dysplasia or gastric cancer in individuals with *H. pylori* infection.

## 1. Introduction

Although the incidence of gastric cancer has declined over the last century, it remains the third leading cause of cancer deaths worldwide; nearly a million new cases of gastric cancer are diagnosed annually, of which approximately two-thirds occur in developing countries [1,2]. The putative mechanism of gastric carcinogenesis involves inflammation and epithelial damage, which lead to gastric precancerous lesions (atrophic gastritis, intestinal metaplasia, and dysplasia) and gastric cancer, in that order [3].

In the USA, gastric cancer was the leading cause of cancer mortality until 1930; however, incidence and mortality declined sharply in the 20th century in the USA [4,5]. This decline led researchers to hypothesize that dietary and environmental factors may be involved in the development of gastric cancer [6]. During the first half of the 20th century, methods of food preservation rapidly transitioned from the use of salt to refrigeration, as the use of refrigeration began and became widespread [4]. As a reduction in gastric cancer incidence and mortality seemed to coincide with a reduction in the use of salt in food preservation, it was hypothesized that salt intake may be involved in the etiology of gastric cancer [4]. The earliest reported evidence of a potential association between salt intake and gastric cancer came from a few animal and ecological studies [7,8,9,10,11]. Since then, a number of cross-sectional, case-control, and cohort studies have investigated the effects of salt intake on progression to precancerous lesions and/or gastric cancer but these studies have mainly used self-reported measures of salt intake and reported a wide range of effect estimates [12,13,14,15,16,17,18,19,20,21,22,23,24,25,26,27,28,29,30,31,32,33,34,35,36,37,38,39,40,41,42,43]. The value of self-reported measures of salt intake may be limited because individuals may not know the amount of salt added to or naturally present in foods [13,44]. Additionally, salt intake estimation from self-reported measures has been reported to have a low correlation with urinary sodium/creatinine ratios obtained from 24-h urine samples, the gold standard for dietary salt intake estimation [44,45]. Hence, most previous studies may not have accurately measured salt intake [13,44]. Although measurement of sodium/creatinine ratios from 24-h urine samples is considered the best way to determine salt intake [44], obtaining a 24-h urine sample is not generally feasible because it requires the study participants to be hospitalized. Estimating urinary sodium/creatinine ratios from spot urine samples, which closely approximates the measures from 24-h urine samples, is a more feasible approach and has greater validity compared to self-reported measures [45].

Hence, the use of urinary sodium/creatinine ratios to estimate dietary salt intake is important to increase the validity of studies estimating the effect of salt intake on the gastric precancerous process. However, there is a paucity of such studies. Further, cohort studies with adequate follow-up whereby salt intake is measured before progression to a precancerous lesion or gastric cancer has occurred are needed. Furthermore, the pathogenesis of gastric cancer is multifactorial [1,2,3]; several factors including *Helicobacter pylori* infection and diet may affect this process simultaneously [1,2,3]. Therefore, the presence of *H. pylori* may modify the effect of dietary salt intake on progression in the gastric precancerous process. To address these gaps, we analyzed data from a cohort study conducted in Colombia, South America [46] to estimate the effect of salt intake at baseline on future progression of gastric precancerous lesions in the entire cohort and within subgroups with and without *H. pylori* infection.

## 2. Results

Of the 399 participants with baseline data for sodium and creatinine, 354 were successfully followed-up at 5 and 309 months for 11 to 12 years. The final eligible analytical sample of 260 participants consisted of those who did not have dysplasia or gastric cancer at baseline who also had complete histologic diagnoses at each time frame, urine sodium/creatinine data at baseline and 5 months, as well as data on weekly fruit and vegetable intake, gender, age, and socioeconomic status for adjustment in the regression models (Appendix A).

At baseline, the average urinary sodium/creatinine ratio was 1.66 mmol/cg (standard deviation: 0.8). Baseline characteristics of the participants are presented in Table 1 by tertiles of sodium/creatinine ratios. Briefly, participants in the highest tertile of urinary sodium/creatinine ratio at baseline were older, of lower socioeconomic status, consumed fewer fruits and vegetables, and had more advanced baseline gastric lesions. All participants were *H. pylori* positive at baseline and all individuals who tested *H. pylori* negative at 5 months had their infection eliminated with treatment.

Among the 260 participants included in the analysis using sodium/creatinine ratio as the exposure, more than 33% (*n* = 85) had some type of progression in the precancerous process after baseline that was documented at the 11–12-year follow-up. There were 28 participants who progressed to atrophic gastritis, 63 participants who progressed to intestinal metaplasia, and 10 participants who progressed to dysplasia or gastric cancer. Some participants progressed to more than one gastric precancerous lesion.

We did not observe a positive association between urinary sodium/creatinine ratio and overall progression in the gastric precancerous process after adjusting for age, gender, socioeconomic status, and fruit and vegetable consumption, nor within subgroups who tested positive or negative for *H. pylori* (Table 2). We also did not observe a positive association between salt intake and progression to intestinal metaplasia. We observed a weak positive association between sodium/creatinine ratios and progression to atrophic gastritis (adjusted RR: 1.21; 95% CI: 0.93–1.57), and a slightly stronger positive association with progression to dysplasia or gastric cancer (adjusted RR: 1.32; 95% CI: 0.96–1.81) (Table 2). Additionally, a stronger association for progression to dysplasia or gastric cancer was observed among those with *H. pylori* infection five months (after treatment) (adjusted RR: 1.53; 95% CI: 1.12–2.09) and those with persistent *H. pylori* infection throughout the follow-up (adjusted RR: 1.49; 95% CI: 1.09–2.05), Table 2. Due to small sample sizes in the *H. pylori* negative group, we were not able to estimate an adjusted risk ratio for progression to dysplasia or gastric cancer. Likewise, we observed urinary sodium/creatinine ratio was positively associated with progression to dysplasia or gastric cancer in the overall cohort comparing the histologic diagnosis from 5-month follow-up (after treatment) to the 11–12 year follow-up (adjusted RR: 1.37, 95% CI: 1.05–1.78), as well as in those who continued to be infected with *H. pylori* five months after treatment (adjusted RR: 1.52; 95% CI: 1.14–2.03), and in those with persistent *H. pylori* infections throughout the study period (adjusted RR: 1.48; 95% CI: 1.11–1.98) (Appendix A). The crude risk ratios are reported in Appendix A.

The association between salt intake, as estimated from frequency of adding salt to foods and from the amount of salt added to foods, and progression to gastric dysplasia or gastric cancer were similar to those estimated from urinary sodium/creatinine ratios, but the observed effects were less precise and tended to be weaker (Appendix A).

## 3. Discussion

In this study, we investigated the effects of salt intake on overall and lesion-specific progression in the gastric precancerous lesions and the ways in which those effects were modified by the continued presence of *H. pylori* infection after treatment. We observed an association between salt intake and progression to advanced lesions (dysplasia and gastric cancer), particularly among those who continued to be infected with *H. pylori* after treatment. We did not observe a positive association between salt intake and progression in the gastric precancerous process overall, and the data were too sparse to calculate an adjusted estimate for the RR for advanced lesions (dysplasia and gastric cancer) among those without *H. pylori* infection.

In our review of the literature, we found 30 (19 case-control, 10 cohort, and one cross-sectional) [12,13,14,15,16,17,18,19,20,21,22,23,24,25,26,27,28,29,30,31,32,33,34,35,36,37,38,39,40,41,42,43] epidemiological studies that estimated the effect of salt intake on gastric precancerous lesions and/or gastric cancer and provided data to estimate the effect of salt intake on gastric precancerous lesions or cancer. This review revealed that some previous studies reported increased risk of advanced lesions (dysplasia or gastric cancer) with increased salt intake [12,13,15,17,19,20,23,27,28,30,31,33,34,37,38,40,41,42,43], consistent with our study, while others did not observe an association [14,16,17,20,21,24,25,26,29,30,31,32,35,36,43]. The differences in the estimates of salt intake on gastric precancerous lesions and/or gastric cancer from previous studies may have been due to potential effect modification by *H. pylori* infection, that were not accounted for in previous studies, but were in our study. Further, inconsistencies between our results and those from some studies may be due to inaccuracies in the measure of salt consumption using self-reports in all but one [13] previous studies. Self-reported measures of salt intake have been reported as less accurate estimates of dietary salt intake compared to urinary sodium/creatinine ratios because urinary sodium/ creatinine ratios are not prone to inaccurate recall and capture salt intake from all sources [44,45,47,48]. Salt intake estimated from self-reported measures only accounts for recalled food items listed in the questionnaires and are dependent on the average salt content of each food item per serving provided by the standard nutrition tables. Sizes of each serving and the amount of salt added while preparing the food items may vary by individuals and this may influence the amount of salt content in the food items. Hence, the salt content of each food item provided by the standard nutrition tables may not be representative across individuals. This may decrease the accuracy of self-reported measures of salt intake. Therefore, in addition to the self-reported measures, we used urinary/sodium creatinine ratios to estimate salt intake in our study. We used spot urine samples to estimate sodium/creatinine ratios because the collection of 24-h urine samples was not feasible as the participants were not hospitalized. However, urinary sodium/creatinine ratios estimated from spot urine samples have been reported to be highly correlated with the measures from the gold-standard 24-h urinary sodium/creatinine ratios and have been considered a “potentially simpler method to estimate [sodium] intake and may be used to replace 24-h urinary [sodium]” [45].

We are only aware of one previous study, a cross-sectional study, by Chen et al. [13] which used urinary sodium/creatinine ratios from spot urine samples to estimate the effect of salt intake on gastric precancerous lesions. As the Chen et al. [13] study was cross-sectional, salt intake and precancerous lesions were both measured at the same point in time, and therefore the results could be affected by temporal ambiguity and protopathic bias. Protopathic bias may result from possible changes in dietary habits after the diagnosis of the gastric precancerous lesions such that the timing of urine sample collection would not reflect salt intake that may have contributed to the development of the precancerous lesions [49]. In the current study, we limited the possibility of protopathic bias by using sodium/creatinine ratios measured before the outcome, progression in the gastric precancerous lesions, occurred. The study by Chen et al. [13] was similar to the current study in that salt intake was found to increase the risk of advanced gastric precancerous lesions (dysplasia), and atrophic gastritis to a lesser extent. Overall, our estimates were weaker than those observed by Chen et al. [13].

A potential mechanism for the effect of *H. pylori* infection on the association between salt intake and gastric cancer has been reported in animal studies. In mice and Mongolian gerbils, high salt diet may exacerbate *H. pylori*-induced gastric cancer [46,50,51]. In mice, this occurs as increased proliferation and elongation of gastric pits, which may also provide more room for *H. pylori* to colonize [51]. Increased colonization of *H. pylori* in the stomachs of the animals may lead to increased gastric inflammation and epithelial damage, which may subsequently lead to gastric precancerous lesions and cancer [50,51,52]. The effect of salt intake on epithelial damage, i.e., mucus depletion, is more prevalent at later stages in the cascade, which may explain the stronger association observed in the current analysis between salt intake and progression to advanced precancerous lesions or cancer, compared to earlier lesions. The presence of a potential interaction between salt intake and *H. pylori* colonization to increase the risk of advanced gastric precancerous lesions/cancer could be the reason behind the observed stronger positive effect among those with *H. pylori* infections.

Our study has some limitations. First, limited power prevented us from assessing statistical interaction between salt intake and *H. pylori* infection to increase gastric association precancerous progression. In our sub-group analyses by *H. pylori* infection status, we observed stronger effects of salt intake on progression to dysplasia or gastric cancer among those with *H. pylori* infection, but the data were too sparse to estimate the adjusted effect for those without the infection. Second, loss to follow-up and incomplete data was substantial over 12 years. However, selection bias due to selective loss to follow-up is unlikely because selection bias would have occurred only if the risk for progression in the gastric precancerous lesions (which would be unknown to the patients and their physicians) influenced loss differently at different levels of salt intake. Finally, we investigated the effect of salt intake on progression mainly to precancerous lesions because only one gastric cancer case occurred in the study cohort at the 12-year follow-up assessment. Gastric precancerous lesions were the surrogate outcomes for gastric cancer. The limitation of using surrogate outcomes is that the effect of the exposure on the final clinical outcome may be different from that observed on the surrogate outcome. This is unlikely to have occurred in our study, particularly where the outcomes were progression to dysplasia or cancer because nearly 65–80% of those that develop dysplasia are likely to progress to gastric cancer [53].

Our study has several strengths. First, as previously mentioned, unlike previous studies, we estimated salt intake with three methods (urinary sodium/creatinine ratios, self-reported frequency of adding salt to food, and table salt intake). Furthermore, we used the average sodium/creatinine ratios from baseline and five months to increase the reliability of urinary sodium/creatinine ratios. Second, to our knowledge, ours is the first study with a follow-up design to investigate modification of the association of salt on the gastric precancerous process by *H. pylori* infection. Only three studies have reported estimates for the effect of salt intake on gastric cancer among those with and without *H. pylori* infection, all of which were case-control in design and used self-reported measures of salt intake [23,28,38]. Third, we used multiple biopsies to reduce misdiagnoses of gastric precancerous lesions and *H. pylori* infections because they tend to occur in patches [54] and can be missed if multiple biopsies are not used. Fourth, unlike previous studies, we adjusted for potential confounders (age, socioeconomic status, and fruit and vegetable intake) identified by directed acyclic graphs. An *e*-value can assist us in evaluating whether our results can likely be attributed to confounding and is defined as “the minimum strength of association, on a risk ratio scale, that an unmeasured confounder would need to have with both [exposure] and outcome to fully explain away a specific [exposure]–outcome association, conditional on the measured covariates” [55]. The *e*-value for our study is 2.43, which means that an unmeasured confounder would have to increase the risk of progression more than two-fold (RR = 2.43) and simultaneously would have to show a 2.43 relative risk for the association between the confounder and salt. Although possible, it is unlikely that such strong associations simultaneously exist. Thus, we likely adjusted adequately for confounding, and our results are not likely due to residual confounding.

## 4. Materials and Methods

### 4.1. Study Population

We analyzed existing data from a prospective cohort study that was originally conducted in Pasto, Colombia in 1993 [52]. The original cohort came from of a 16-week clinical trial of five treatment regimens: (a) metronidazole, amoxicillin, and bismuth subsalicylate for two weeks followed by bismuth subsalicylate alone for the next 14 weeks, (b) calcium carbonate for 16 weeks, (c) treatment regimens (a) and (b), (d) tetracycline for 16 weeks or (e) placebo for 16 weeks [52]. The treatment was followed by a short-term follow-up assessment (five months after baseline) and a long-term follow-up assessment (11–12 years after baseline). The purpose of the original cohort study was to investigate the efficacies of the treatment regimens in reducing gastric inflammation and epithelial damage among participants infected with *H. pylori* [52]. Study participants were residents of Pasto, the capital of the department of Nariño, Colombia [52], who were recruited mainly through a community public service radio announcement, with only a few recruited through referrals made by local physicians [52]. Eligible participants at baseline were ages 18–65 years, had symptoms consistent with non-ulcer dyspepsia, *H. pylori* infection, resided within the city limits of Pasto, Colombia with no intentions to move out of Pasto, and provided informed consent [52]. Pregnant women and those with allergies, heavy alcohol use, use of medication that would interfere with the trial medications, or diseases, other than mild hypertension controlled with medication, were excluded [52]. Participants who were diagnosed with dysplasia or gastric cancer at baseline were also excluded from the current analyses. This cohort study was approved by the ethics committee at the Universidad del Valle in Cali, Colombia (“Estudio de una cohort de gastritis asociada de *H. pylori*” (372)).

### 4.2. Measurement of Dietary Salt Intake

We measured salt intake in three ways: urinary sodium/creatinine ratio (the ratio accounts for urine volume and dilution), self-reported frequency of adding salt to food, and total salt added to food. Urinary sodium/creatinine ratios capture salt intake from food and other sources and are reported to measure salt intake more accurately than self-reported salt intake from food frequency questionnaires [44,45,47,48].

At baseline, five-month follow-up, and 11–12-year follow-up, first morning urine samples were collected after a fast that started at midnight. We estimated urinary sodium/creatinine ratios from the measured concentrations of sodium (mmol/L) and creatinine (mg/dL) by dividing urinary sodium in mmol/L by urinary creatinine in g/L. For ease of interpretation, we converted sodium/creatinine ratios from mmol/g to mmol/cg. For our analysis, we used the average of sodium/creatinine ratios measured at baseline and five months as one of the measures of salt intake because studies have recommended using an average of multiple measures to increase reliability and validity [44]. In our data, we observed a strong correlation between baseline and five-month sodium/creatinine ratios (adjusted *R*^2^ = 0.92), which allowed for averaging across the two measures to increase reliability.

At baseline, participants self-reported the frequency with which they (or whomever cooks in the household) added salt to food, which was categorized as rarely or never, occasionally, and always or frequently. Additionally, we asked participants to add the same quantity of salt to a container as was added to food. At follow-up visits, the research team weighed and recorded the amount of salt in the container, which was then categorized into quintiles.

### 4.3. Measurement of Progression in the Gastric Precancerous Process and *Helicobacter pylori* Infection

According to Correa’s precancerous cascade, non-atrophic gastritis is followed by atrophic gastritis, intestinal metaplasia, dysplasia, and gastric cancer in that order [3]. The primary outcome examined in the current study was an overall progression in the gastric precancerous process, defined as any sequential progression in the gastric precancerous cascade [3]. Histological diagnoses at baseline and follow-up were used to determine progression in the gastric precancerous lesions from gastric biopsies collected at baseline, five-month follow-up, and 11–12-year follow-up [52]. The biopsies were embedded in paraffin, sectioned, and stained with hematoxylin-eosin [52]. The updated Sydney system was used to classify and grade each biopsy for precancerous lesions [52,54,56]. The biopsies were also stained with modified Steiner stain and scored for the density of *H. pylori* infection [47]. For all samples analyzed, three pathologists provided independent histological reports. Any discrepancies in the histological reports were resolved by discussions among the study pathologists until a consensus was reached.

### 4.4. Statistical Analysis

We conducted Poisson regression modelling using all trial participants with complete data to estimate risk ratios and their 95% confidence intervals (CI) for the association between salt intake and: (a) Overall progression in the gastric precancerous process and (b) progression to each gastric precancerous lesion. We used Poisson regression because, unlike logistic regression, it does not overestimate risk ratios when the outcomes are common. In this study, the primary outcome—progression in the precancerous process—was relatively common (33%). All models were adjusted for baseline measures identified as potential confounders by directed acyclic graphs [57] and the existing literature showing socioeconomic status [58,59], fruit and vegetable intake [58,60], and age associated with both salt intake and progression in the gastric precancerous process. We also adjusted for gender, a risk factor for gastric cancer, as salt intake may be associated with gender. Directed acyclic graphs are causal diagrams that help visualize the relations between exposures, outcomes, and covariates together [57]. In all the multivariate analyses, the sodium/creatinine ratio was included in the models as a continuous variable. All the outcomes of interest (progression) were dichotomous. The group at risk for the outcome who progressed to the specific lesion(s) were compared the group at risk for the outcome who did not progress to that specific lesion(s). For example, those at risk for atrophic gastritis included only those who were diagnosed with no gastritis or non-atrophic gastritis at baseline.

To determine whether *H. pylori* infection status five months after treatment modified the association between salt intake and progression in the gastric precancerous process, we conducted subgroup analyses among those with and without *H. pylori* infection. We repeated these analyses for those with persistent *H. pylori* infections at every time point until the 11–12-year follow-up assessment as *H. pylori* treatment may have been used after the initial trial, and some participants who had their infection eliminated after treatment were reinfected during follow-up. We also used *H. pylori* infection status at five months for the stratified analyses to account for possible treatment-related changes in *H. pylori* infection status during the randomized clinical trial (from baseline to five months). All analyses were conducted using SAS 9.4 (SAS institute, Cary, NC, USA).

## 5. Conclusions

To our knowledge, all previous studies (except one cross-sectional study) investigated the association between salt intake and gastric precancerous lesions and/or gastric cancer using self-reported measures of salt intake. We conducted a study with a follow-up design which estimated the association between salt intake measured by urinary sodium/creatinine ratios and progression in the gastric precancerous process, and possible modification of this association by *H. pylori* infection at each stage in the gastric precancerous process. In this study, we observed that salt intake may increase progression to advanced gastric precancerous lesions (gastric dysplasia) or gastric cancer among those who continued to be infected with *H. pylori* after treatment or who had persistent *H. pylori* infection throughout the study period. Future studies that use urinary sodium/creatinine ratios to measure salt intake with larger sample sizes are needed to confirm these results and to further understand the effect of salt intake on gastric precancerous lesions and gastric cancer.

## Figures and Tables

**Table 1 cancers-11-00467-t001:** Baseline characteristics by tertiles of sodium/creatinine ratios.

Characteristics		Overall *n* = 260	Urinary Sodium/Creatinine Ratio		
			Tertile 1 *n* = 86 Range = 0.210–1.332	Tertile 2 *n* = 87 Range = 1.333–2.046	Tertile 3 *n* = 87 Range = 2.058–7.417
Age ^a^		43.8 (11.5)	41.7 (11.1)	42.6 (11.7)	47.9 (10.8)
Sex (%females) ^b^		66.1	71.7	54.6	70.2
Car ownership (%) ^b^		29.2	33.6	37.5	16.0
Education (years) ^a^		6.9 (4.8)	7.9 (4.7)	7.1 (5.4)	5.3 (3.8)
Fruit and vegetable servings per week ^a^		66.9 (26.3)	69.4 (28.8)	69.0 (24.7)	64.9 (25.6)
Baseline diagnosis (%) ^b^	No gastritis	2.4	6.2	-	-
	Non-atrophic gastritis	31.2	38.9	26.1	26.6
	Atrophic gastritis	35.3	33.6	37.5	35.1
	Intestinal metaplasia	28.5	20.4	34.1	33.0
	Dysplasia and gastric cancer	2.7	0.9	2.3	5.3

^a^ Mean (standard deviation); ^b^ percentage of the overall or tertiles of sodium/creatinine ratio.

**Table 2 cancers-11-00467-t002:** Risk ratios for the association between salt intake (defined by the average of the baseline and five-month measures of urinary sodium/creatinine ratio) and progression in the gastric precancerous process (11–12 years compared to baseline).

Outcomes	Overall		*H. pylori* Positive at Five Months		*H. pylori* Negative at Five Months		Persistent *H. pylori* (Positive at Five Months and 11–12 years)	
	*n* with: *n* without Outcome	Risk Ratio (CI) *p*-Value ^a^	*n* with: *n* without Outcome	Risk Ratio (CI) *p*-Value ^a ^	*n* with: *n* without Outcome	Risk Ratio (CI) *p*-Value ^a ^	*n* with: *n* without Outcome	Risk Ratio (CI) *p*-Value ^a ^
Overall progression in the gastric precancerous process Total *n* at risk = 260 ^c^	85:175	0.94 (0.76–1.15) *p* = 0.52	72:144	0.95 (0.76–1.19) *p* = 0.68	13:31	0.80 (0.43–1.48) *p* = 0.48	64:121	0.96 (0.77–1.20) *p* = 0.72
Incident progression to atrophic gastritis Total *n* at risk = 73	28:45	1.21 (0.93–1.57) *p* = 0.14	23:37	1.13 (0.85–1.51) *p* = 0.38	5:8	1.92 (0.67–5.48) *p* = 0.22	20:29	1.07 (0.77–1.49) *p* = 0.69
Incident progression to intestinal metaplasia Total *n* at risk = 175	63:112	0.89 (0.71–1.11) *p* = 0.29	56:89	0.91 (0.73–1.15) *p* = 0.45	7:23	0.81 (0.31–2.16) *p* = 0.68	51:73	0.97 (0.78–1.20) *p* = 0.75
Incident progression to dysplasia or gastric cancer Total *n* at risk = 253	10:243	1.32 (0.96–1.81) *p* = 0.09	8:203	1.53 (1.12–2.09) *p* = 0.007	2:40	^b^	8:172	1.49 (1.09–2.05) *p* = 0.01

^a ^ Adjusted for age, gender, car ownership, and fruit and vegetable intake. ^b^ Missing risk ratio due to sparse data. ^c^ Total at risk overall is less than the sum of the total at risk for each specific histological diagnosis since participants can be at risk for more than one outcome at baseline. CI: confidence interval, *H. pylori*: *Helicobacter pylori*

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
