# Peer review of "Association between Dietary Salt Intake and Progression in the Gastric Precancerous Process"

_cancers, 2019, doi:10.3390/cancers11040467_

Reviewer 1 Report

Thapa et al. have conducted a longitudinal study to assess the association of dietary salt intake with progression in the gastric cancer cascade with stratification by Helicobacter pylori infection. The aim of this study is highly justified, and would be a sound contribution to the literature. However, while the study design and methods for assessment of dietary salt intake seem adequate, the major limitation of this manuscript is the presentation of statistical methods and study results. Section-by-section comments are given below.

Abstract:

1)      In addition to describing the RR in H. pylori positive study participants, display of the respective RR in H. pylori negative individuals could help justify the overall conclusion.

Introduction:

1)      L. 88 mentions the aim of stratification by H. pylori. A description of H. pylori’s role in gastric carcinogenesis would be an important information to justify this aim.

2)      Information in ll. 50-63 should be summarized to improve the flow of the introduction.

Materials and methods:

1)      Study population: please describe the study population more thoroughly, including: a) the initial purpose of the clinical trial; b) the number of participants initially recruited for the cohort; c) how many of these were then included in the presented study; d) as well as from which trial arm were the study participants recruited? (a flow-chart could help present this information).

2)      Measurement of salt intake: In ll. 301-304 the authors describe that the average of the results from urine at baseline and 5 months was used as baseline value. Please provide information on the variation in sodium/creatinine between the two time points and how this justifies to combine the two measurements.

3)      Assessment of outcome: Please provide information on the reference/control group in your analysis.

4)      Statistical analysis: a) how do a priori defined confounders relate to exposure and outcome in your study? Did adjustment for these factors alter the estimate? Why was sex not considered as potential confounder?; b) was the trial arm considered to modify the association?; c) It is unclear from the described methods how salt intake is analyzed in the model, i.e. on a continuous scale or in tertiles (as presented in table 1); d) the authors describe in ll.340-347 the stratified analysis by H. pylori infection at different time points; did the authors consider to assess an interaction between salt intake and H. pylori infection?

Results:

1)      Please include n’s in table 1.

2)      Please revise table 2 and 3 to make presentation of n’s in each group and resulting RR more clear: a) give n’s for reference groups; b) n’s in individual progression groups do not add up to the overall progression group as well as the stratified groups do not add up to the overall group; c) the 95% CIs for the progression to dysplasia/cancer group seem comparably narrow to the small sample size in this group (n=10); d) please provide information on salt intake in the respective groups to allow the reader to understand the resulting RR

3)      Table 3: why are the overall n’s smaller in table 3 than table 2?

4)      The authors only shortly mention the results for dietary salt intake obtained by questionnaire data in the results. A supplementary table showing the respective data would be helpful to understand the described results.

Discussion:

1)      It is appreciated that the authors describe in detail potential residual confounding, however, to improve the flow of the discussion this part should be more concise.

Author Response

We would like to thank the reviewers for their thoughtful comments and suggested edits to our manuscript. Below is a point-by-by response to the reviewers’ comments (the reviewers’ comments are in quotations).

Reviewer 1

“Thapa et al. have conducted a longitudinal study to assess the association of dietary salt intake with progression in the gastric cancer cascade with stratification by Helicobacter pylori infection. The aim of this study is highly justified, and would be a sound contribution to the literature. However, while the study design and methods for assessment of dietary salt intake seem adequate, the major limitation of this manuscript is the presentation of statistical methods and study results. Section-by-section comments are given below.”

Reviewer: “Abstract: 1) In addition to describing the RR in H. pylori positive study participants, display of the respective RR in H. pylori negative individuals could help justify the overall conclusion.”

Response: We added and modified the following sentence in the abstract per reviewer suggestion: “We did not observe an association between salt intake and overall progression in the gastric precancerous process among participants without H pylori infection (adjusted RR: 0.80 (0.43, 1.48). Conclusion: Salt intake may increase the risk of gastric dysplasia or gastric cancer in individuals with H. pylori infection.”

Reviewer: “Introduction: 1) L. 88 mentions the aim of stratification by H. pylori. A description of H. pylori’s role in gastric carcinogenesis would be an important information to justify this aim.”

Response: We have added the following sentence in the last paragraph of the “Introduction” to justify the subgroup analyses by H. pylori infection status: “Furthermore, the pathogenesis of gastric cancer is multifactorial [1, 2, 3]; several factors including H. pylori infection and diet may affect this process simultaneously [1, 2, 3]. Therefore, the presence of H pylori may modify the effect of dietary salt intake on progression in the gastric precancerous process.”

Reviewer: “2) Information in ll. 50-63 should be summarized to improve the flow of the introduction.”

Response: We have edited the introduction to improve the flow and make it more concise.

Reviewer: “Materials and methods: 1) Study population: please describe the study population more thoroughly, including: a) the initial purpose of the clinical trial;”

Response: We have now added information including the original purpose of the cohort study (1st paragraph-Materials and Methods).

Reviewer: “b) the number of participants initially recruited for the cohort; c) how many of these were then included in the presented study; d) as well as from which trial arm were the study participants recruited? (a flow-chart could help present this information).”

Response: We have now included a flowchart in our Supplementary Material I for the sample derivation of the participants included in the present study.  We also clarify that our analyses included all trial participants with complete data in the first sentence of the Statistical Analysis section.

Reviewer: “2) Measurement of salt intake: In ll. 301-304 the authors describe that the average of the results from urine at baseline and 5 months was used as baseline value. Please provide information on the variation in sodium/creatinine between the two time points and how this justifies to combine the two measurements.”

Response: We used the calculated adjusted R2 to estimate the correlation between baseline and 5-month sodium/creatinine ratio. The adjusted R2 was 0.92 indicating a high correlation between the two measures. Furthermore, the use of averages for multiple measures has been reported to increase reliability of overall estimates. We have now included this information in the methods under “Measurement of Dietary Salt Intake” (2nd paragraph, last sentence) to justify our use of an average measure.

Reviewer: “3) Assessment of outcome: Please provide information on the reference/control group in your analysis.”

Response:  We compared those who progressed in the gastric precancerous cascade to those who did not progress.  For each precancerous lesion, we used those who were at risk for progression to the lesion who progressed compared to those who were at risk for progression who did not progress.  We have added sentences in the Statistical Analysis section to clarify this.

Reviewer: “4) Statistical analysis: a) how do a priori defined confounders relate to exposure and outcome in your study? Did adjustment for these factors alter the estimate? Why was sex not considered as potential confounder?; b) was the trial arm considered to modify the association?;”

Response: Adjustment of a priori confounders changed some of the estimates, especially for progression to dysplasia/cancer.  For example, the Crude RR was 1.35 (CI: 0.95, 1.91) compared to the Adjusted RR: 1.53 (CI: 1.12, 2.09) for progression to dysplasia or gastric cancer among H pylori positive participants. It is recommended by many epidemiologists that potential confounders to be controlled for be identified using a priori knowledge (if it exists) from the literature and common sense (using Directed Acyclic Graphs or DAGs) rather than using the current study data (using a ≥10% difference rule).  Therefore, we did not assess the association between each confounder and the exposure and outcome in the data [Greenland S, Pearl J, Robins JM. Causal diagrams for epidemiologic research. Epidemiology. 1999;10(1):37-48.]

We have reanalyzed the data adjusting further for gender, and we only observed very slight changes in our estimates.  Table 2 and the text have also been edited (including the Statistical Analysis section) to reflect this new analysis with added adjustment for gender. Unfortunately, with the addition of adjustment for gender, we can no longer estimate the RR for progression to dysplasia or gastric cancer among those H pylori negative at 5 months, as the data with adjustment was too sparse.  We have commented on this in the Results and Discussion section.

The trial arm was not considered as a confounder in the analyses since treatment regimens were assigned using randomization and therefore it is unlikely they would be associated with the exposure (salt intake). Nevertheless, we compared the treatment arms, and observed them to have a similar mean baseline value for sodium/creatinine.  Therefore, we did not control for treatment arm.

Reviewer: “c) It is unclear from the described methods how salt intake is analyzed in the model, i.e. on a continuous scale or in tertiles (as presented in table 1);”

Response:  Tertiles of salt intake were presented in Table 1 only so that the readers could visualize the baseline characteristics by various levels of salt intake. We used salt intake as a continuous variable in the multivariate analyses. We have added additional sentences under “Statistical Analysis” (1st paragraph) to clarity this.

Reviewer: “d) the authors describe in ll.340-347 the stratified analysis by H. pylori infection at different time points; did the authors consider to assess an interaction between salt intake and H. pylori infection?”

Response: Our study lacked power to observe statistically interactive effects. We have now added this as a limitation in our “Discussion” (5th paragraph under “discussion”).

Reviewer: “Results: 1) Please include n’s in table 1.”

Response: We have now included the sample size for participants in Table 1.

Reviewer: “2) Please revise table 2 and 3 to make presentation of n’s in each group and resulting RR more clear: a) give n’s for reference groups; b) n’s in individual progression groups do not add up to the overall progression group as well as the stratified groups do not add up to the overall group;”

Response: We have now included the number of participants who experienced the outcome, and the number who did not experience the outcome (N with / N without the outcome) for each outcome in Table 2. The sample size overall does not add up to the sum of the individual progression groups since participants can be at risk for more than one outcome.  For example, participants diagnosed with non-atrophic gastritis at baseline are at risk to progress to atrophic gastritis, intestinal metaplasia, dysplasia and gastric cancer. So, if participant X progressed from non-atrophic gastritis to dysplasia, they would be classified as having progressed to atrophy, intestinal metaplasia and dysplasia.  We have added a new footnote to Table 2 to clarify this. There was an error in the sample sizes for the subgroups with and without H pylori, which we have now corrected.

Reviewer: “c) the 95% CIs for the progression to dysplasia/cancer group seem comparably narrow to the small sample size in this group (n=10); d) please provide information on salt intake in the respective groups to allow the reader to understand the resulting RR.”

Response: The overall analytic sample size used for the outcome progression to dysplasia or gastric cancer group was 253 for Table 2; 10 of these experienced progression to dysplasia or gastric cancer. The higher overall sample size at risk for dysplasia (N=253) compared to the sample size for atrophic gastritis (N=73) or intestinal metaplasia (N=175) may have contributed to more narrow confidence interval despite only 10 participants experiencing progression to dysplasia or gastric cancer in the analytic sample. We have provided the sample size of those who progressed and those who did not progress for each outcome to clarify the differences in sample sizes used (at risk for the outcome) for each analysis.

Reviewer: “3) Table 3: why are the overall n’s smaller in table 3 than table 2?”

Response: For our original Table 2 we looked at progression from baseline to 11-12 years whereas for our original Table 3 we looked at progression (histology) from 5 months to 11-12 years. Histological assessments at baseline and 5-months were not done on the same number of participants and therefore, there are differences in the number in each group. Furthermore, the diagnosis at baseline may have changed at 5 months, especially in the atrophic gastritis group where regression from atrophic gastritis to non-atrophic gastritis was common soon after treatment. Although the estimates for Table 2 and Table 3 are similar, this appears to be confusing for the reader, and therefore we have omitted our original Table 3 from the manuscript and include it only in the supplemental material.

Reviewer: “4) The authors only shortly mention the results for dietary salt intake obtained by questionnaire data in the results. A supplementary table showing the respective data would be helpful to understand the described results.”

Response: We have now included additional supplementary tables (Supplemental Material III and IV) for salt intake measured using questionnaires and salt measured by weight added to food.

Reviewer: “Discussion: 1) It is appreciated that the authors describe in detail potential residual confounding, however, to improve the flow of the discussion this part should be more concise.”

Response: We have slightly shorted this part of the discussion section regarding the topic of residual confounding.

Reviewer 2 Report

This is a well-written manuscript examining the association between dietary salt intake and progression in the gastric precancerous process. The authors also tested the associations among those with persistent H.pylori infection and among those who did not have H.pylori infection.

Introduction:

    1.       Introduction overall is well written but could be shortened to improve the manuscript. For e.g Line 77-82 could be restated to avoid redundancy and present a precise rationale.

Methods:

    2.       Study population:

        a.       Although it seems obvious, it is helpful for the reader to mention that these treatments are for H.pylori infection.

        b.      The authors also did not mention whether the current analyses included participants from both experimental and control arms or if any of them were excluded. A simple sample derivation flowchart would be very helpful even as a supplemental file.

        c.      Were all participants diagnosed with H.pylori at baseline? If not, the statements in the manuscript should be clear whether individuals in the H.pylori negative group never had infection or were successfully treated following the medication regimens.

   3.       Line  301: The authors mentioned conversion of units to mmol/cg but in Line 99 of  “Results” section, they presented mmol/mg. This has to be consistent.

   4.       Statistical analysis:

         a.    The authors did not mention to what value was α set a-priori. Was it 0.05 or 0.1?

         b.      The authors did not mention the software that they used for analyses.

Results:

    5.       Table 1: Please mention the min-max or median (IQR) for the tertiles of urinary sodium/creatinine ratio

    6.       Tables 2 and 3: Be consistent in denoting ‘n’ or ‘N’ for number of individuals and mention the number of events/cases/total for that group in each ‘n’ column. For e.g. in Table 2, first row, did 85 individuals out of 260 progress in gastric precancerous progress? What about the other strata

Discussion:

    7.       Line 247: Extra space before beginning of sentence

Author Response

We would like to thank the reviewers for their thoughtful comments and suggested edits to our manuscript. Below is a point-by-by response to the reviewers’ comments (the reviewers’ comments are in quotations).

Reviewer 2

“This is a well-written manuscript examining the association between dietary salt intake and progression in the gastric precancerous process. The authors also tested the associations among those with persistent H.pylori infection and among those who did not have H.pylori infection.”

Reviewer: “Introduction: 1. Introduction overall is well written but could be shortened to improve the manuscript. For e.g Line 77-82 could be restated to avoid redundancy and present a precise rationale.”

Response: We have edited the introduction as suggested by the reviewers to make it more concise.

Reviewer: “Methods: 2.  Study population: a.  Although it seems obvious, it is helpful for the reader to mention that these treatments are for H. pylori infection.”

Response: We have added additional sentences in the first paragraph of “Materials and Methods” to specify the objective of the original trial.

Reviewer: “b.  The authors also did not mention whether the current analyses included participants from both experimental and control arms or if any of them were excluded. A simple sample derivation flowchart would be very helpful even as a supplemental file.”

Response: We have now added a sample derivation flowchart as Supplementary Material I and we added clarification to the first sentence in the analysis section to clarify that participants from all treatment arms were included in the analyses.

Reviewer: “c.  Were all participants diagnosed with H.pylori at baseline? If not, the statements in the manuscript should be clear whether individuals in the H.pylori negative group never had infection or were successfully treated following the medication regimens.”

Response: All participants included in this analysis were H. pylori positive subjects at baseline. We have added a sentence to the “Materials and Methods” section under “Study Population,” and in the second paragraph of the “Results” section to clarify this.

Reviewer: “3. Line  301: The authors mentioned conversion of units to mmol/cg but in Line 99 of “Results” section, they presented mmol/mg. This has to be consistent.”

Response: We have corrected the typographical error from line 99 of the “Results” section in our original manuscript to clarify that the units for the sodium/creatinine ratio were mmol/cg.

Reviewer: “4. Statistical analysis: a. The authors did not mention to what value was α set a-priori. Was it 0.05 or 0.1?”

Response: We now clarify that 95% Confidence Intervals were used (first sentence of the “Statistical Analysis” section) (which corresponds to an α of 0.05). 

We also want to note that our interpretation of the results was not determined by whether or not the p-value crossed the threshold of 0.05.  Instead, we focused on the measure of effect (the Risk Ratios) and the range of RRs in the confidence intervals. Many epidemiologists, research methodologist, statisticians and others have, over the last several decades, pointed out that the use of a p-value cutpoint to interpret a study’s results is misleading and uninformative.  However, it wasn’t until 2016 that the American Statistical Association (ASA) finally took a strong position and wrote a formal statement against the use of p-values and statistical testing to draw conclusions (Ronald L. Wasserstein & Nicole A. Lazar (2016) The ASA's Statement on p-Values: Context, Process, and Purpose, The American Statistician, 70:2, 129-133, DOI: 10.1080/00031305.2016.1154108). Highlighting the rarity and importance of this statement, this educational policy statement was one of the few made by the ASA over the last 150 years.  As pointed out by the American Statistical Association in their Statement, “let us be clear. Nothing in the ASA statement is new. Statisticians and others have been sounding the alarm about these matters for decades, to little avail. We hoped that a statement from the world’s largest professional association of statisticians would open a fresh discussion and draw renewed and vigorous attention to changing the practice of science with regards to the use of statistical inference.”  Consistent with recommendations made by epidemiologists and statisticians who wrote the ASA statement, we focus on the estimated measures of effect (the Risk Ratios).

Reviewer: “b. The authors did not mention the software that they used for analyses.”

Response: We used SAS 9.4 for all analyses. We have now added this information in the last sentence of “Statistical Analysis” section.

Reviewer: “5 Results:. Table 1: Please mention the min-max or median (IQR) for the tertiles of urinary sodium/creatinine ratio”

Response: We have now included the minimum and maximum range for each tertile of urinary sodium/creatinine ratio in Table 1.

Reviewer: “6.  Tables 2 and 3: Be consistent in denoting ‘n’ or ‘N’ for number of individuals and mention the number of events/cases/total for that group in each ‘n’ column. For e.g. in Table 2, first row, did 85 individuals out of 260 progress in gastric precancerous progress? What about the other strata?”

Response: We now consistently denote the sample size as ‘N’ instead of ‘n’.  The reviewer is correct that 85 out of 260 individuals overall progressed in the gastric precancerous process over the 11-12 years of follow-up. We also now report the total N at risk (e.g. 260 overall) for the outcome (progression in the gastric precancerous process) overall and for each stratum, as well as the N with the outcome (N with the outcome progression overall = 85) and the N without the outcome progress (N who did not progress or in other words without the outcome overall = 175).  We hope this makes Table 2 clearer for the readers.   

Reviewer: “7. Discussion: Line 247: Extra space before beginning of sentence”

Response: We have removed the extra line before the beginning of this sentence.

Reviewer 3 Report

-In the abstract the authors reported “…..the effect of salt intake on overall progression to precancerous lesions and to atrophic gastritis, intestinal metaplasia, and dysplasia or gastric cancer”. Since these steps (in particular intestinal metaplasia and dysplasia) are precancerous lesions I do not understand the difference in the two parts of this sentence.

-When the authors report Helicobacter pylori infection strata, for the readers it is better to simplify this sentence.

-In the introduction should be highlighted that the pathogenesis of gastric cancer is multifactorial and the role of salt should be considered in a context of multiple causative agents. The part of  Introduction dedicated to the description of studies on salt should be shortened.

-The section METHODS should be put in the correct position to better understand this work. The section 2 after Introduction (1) is RESULTS and 4 is MATERIALS AND METHODS.

-In Table 1 it is unclear the prevalence of Helicobacter pylori infection among the considered groups.

-In Table 2 the sum of the several groups is not correct (for example: overall progression 260; H.pylori positive at 5 months 216, H.pylori negative at 5 months 45; hence 261, the same in line 2 etc).

Author Response

We would like to thank the reviewers for their thoughtful comments and suggested edits to our manuscript. Below is a point-by-by response to the reviewers’ comments (the reviewers’ comments are in quotations).

Reviewer 3

Reviewer: “In the abstract the authors reported “…..the effect of salt intake on overall progression to precancerous lesions and to atrophic gastritis, intestinal metaplasia, and dysplasia or gastric cancer”. Since these steps (in particular intestinal metaplasia and dysplasia) are precancerous lesions I do not understand the difference in the two parts of this sentence.”

Response: We agree that this sentence was confusing. We have rephrased this sentence to now read as follows, “Using data from a prospective cohort study, we investigated the effect of salt intake on progression to gastric precancerous lesions.”

Reviewer: “When the authors report Helicobacter pylori infection strata, for the readers it is better to simplify this sentence.”

Response: We have changed Helicobacter pylori infection strata to “with and without H. pylori infection” throughout the manuscript

Reviewer: “In the introduction should be highlighted that the pathogenesis of gastric cancer is multifactorial and the role of salt should be considered in a context of multiple causative agents.”

Response: We have added 2 additional sentences in the last paragraph of the “Introduction” to emphasize this point.

Reviewer: “The part of Introduction dedicated to the description of studies on salt should be shortened.”

Response: We have shortened the description of previous studies in the “Introduction” and the “Discussion” section and have removed Table 4.

Reviewer: “The section METHODS should be put in the correct position to better understand this work. The section 2 after Introduction (1) is RESULTS and 4 is MATERIALS AND METHODS.”

Response: Unfortunately, we are unable to change the order of the sections, as this is the order specified by the journal.

Reviewer: “In Table 1 it is unclear the prevalence of Helicobacter pylori infection among the considered groups.”

Response: We have now clarified that all participants included in this analysis were H pylori positive at baseline in both the “Results” and “Material and Methods” sections.

Reviewer: “In Table 2 the sum of the several groups is not correct (for example: overall progression 260; H.pylori positive at 5 months 216, H.pylori negative at 5 months 45; hence 261, the same in line 2 etc).”

Response: We have now edited any errors in Table 2.

Round  2

Reviewer 1 Report

I thank the authors for the thorough and detailed response to the review. I only have a few minor suggestions:

- Please revise sentences in ll. 43-45; 69; 161-162; and 173-174 since there seem to be words/parts of the sentence missing.

- Supplementary materials V and VI are never mentioned in the manuscript text. If these are for publication purposes please refer to them.

Author Response

Reviewer- “Please revise sentences in ll. 43-45; 69; 161-162; and 173-174 since there seem to be words/parts of the sentence missing.”

Response –1) We have removed the redundant words in lines 43-45. 2) We have shortened the sentence on line 69 to make it more readable. 3) We have replaced the words “in the” with “by” in lines 161-162 (new line-163. 4) On line 174 (new line-175), we have added the word “effect.”

Reviewer- “Supplementary materials V and VI are never mentioned in the manuscript text. If these are for publication purposes, please refer to them.”

Response- We have added a sentence to the last part of paragraph 4 of the “Results” section mentioning Supplementary Materials V and VI.